# Long Non-Coding RNAs Might Regulate Phenotypic Switch of Vascular Smooth Muscle Cells Acting as ceRNA: Implications for In-Stent Restenosis

**DOI:** 10.3390/ijms23063074

**Published:** 2022-03-12

**Authors:** Alberto Arencibia, Fernando Lanas, Luis A. Salazar

**Affiliations:** 1Center of Molecular Biology and Pharmacogenetics, Department of Basic Sciences, Faculty of Medicine, Universidad de La Frontera, Temuco 4811230, Chile; albaren1978@gmail.com; 2Department of Internal Medicine, Faculty of Medicine, Universidad de La Frontera, Temuco 4811230, Chile; fernando.lanas@ufrontera.cl

**Keywords:** long non-coding RNA, competing endogenous RNA, in-stent restenosis, post transcriptional regulation, epigenetics, transcriptomics, in vitro cellular model

## Abstract

Coronary in-stent restenosis is a late complication of angioplasty. It is a multifactorial process that involves vascular smooth muscle cells (VSMCs), endothelial cells, and inflammatory and genetic factors. In this study, the transcriptomic landscape of VSMCs’ phenotypic switch process was assessed under stimuli resembling stent injury. Co-cultured contractile VSMCs and endothelial cells were exposed to a bare metal stent and platelet-derived growth factor (PDGF-BB) 20 ng/mL. Migratory capacity (wound healing assay), proliferative capacity, and cell cycle analysis of the VSMCs were performed. RNAseq analysis of contractile vs. proliferative VSMCs was performed. Gene differential expression (DE), identification of new long non-coding RNA candidates (lncRNAs), gene ontology (GO), and pathway enrichment (KEGG) were analyzed. A competing endogenous RNA network was constructed, and significant lncRNA–miRNA–mRNA axes were selected. VSMCs exposed to “stent injury” conditions showed morphologic changes, with proliferative and migratory capacities progressing from G0-G1 cell cycle phase to S and G2-M. RNAseq analysis showed DE of 1099, 509 and 64 differentially expressed mRNAs, lncRNAs, and miRNAs, respectively. GO analysis of DE genes showed significant enrichment in collagen and extracellular matrix organization, regulation of smooth muscle cell proliferation, and collagen biosynthetic process. The main upregulated nodes in the lncRNA-mediated ceRNA network were PVT1 and HIF1-AS2, with downregulation of ACTA2-AS1 and MIR663AHG. The PVT1 ceRNA axis appears to be an attractive target for in-stent restenosis diagnosis and treatment.

## 1. Introduction

Coronary in-stent restenosis (SR) continues to challenge the scientific community, despite the emergence of new biomedical materials [1] and its association with medications (drug eluting stents) (DES) [2]. SR is a multifactorial process that involves the active participation of vascular smooth muscle cells (VSMCs) and endothelial cells as well as inflammatory and genetic factors [3]. VSMCs have great plasticity, with the ability to modify their synthetic machinery and cyto-architecture in response to various stimuli [4].

VSMCs are typically found in a quiescent state as well-differentiated cells with a functional contractile apparatus, requiring little protein turnover [5]. However, in pathological situations such as stent induced stress their molecular mechanisms are triggered, and VSMCs thereby transform into cells with a high proliferative and migratory capacity that secrete extracellular matrix proteins [6]. This phenotypic switch is the therapeutic target of DES via drugs that inhibit the cell cycle to control the expansion of VSMCs and the formation of neo-intima [7].

Understanding of lncRNA has evolved from being considered “genomic waste” into a position of paramount importance. Next-generation sequencing technologies have allowed annotation and characterization of a number of lncRNAs that exceeds by a large amount the number of protein-coding RNAs. LncRNAs are transcripts of more than 200 nucleotides (with the same posttranscriptional modifications as mRNAs, i.e., 5’-capping, splicing, poly A tail) that do not encode for proteins [8].

Recent studies in the cardiovascular field point to lncRNAs as a key element in cell regulation under physiological and pathological conditions, with special emphasis on cardiac development [9]. The tissue specificity of lncRNA unveils its potential as a disease biomarker [10]. Large-scale genomic association studies (GWAS) determined that polymorphisms in non-coding regions of DNA on chromosome 9p were significantly associated with the risk of acute myocardial infarction; at this locus is the antisense noncoding RNA in the INK4 locus (ANRIL) [11]. Another lncRNA, MIAT (myocardial infarction-associated transcript), has been associated with the risk of myocardial infarction [12].

Transcriptome sequencing of human coronary artery smooth muscle cell (HCASMC) has allowed the identification of 31 lncRNAs not previously described. Several of these lncRNAs showed a preferential expression in vascular cells absent in other cell lines, reaffirming the cellular specificity of these transcripts. A candidate (SENCR) with high expression in vascular smooth muscle and endothelial cells was selected for transfection studies. Its knockdown caused a marked decrease in the proliferation and migration of HCASMC, with a phenotype change from contractile to secretory [13].

An in vitro mouse model showed that lncRNA-p21, a p-53 induced lncRNA, controls the proliferation of vascular smooth muscle in atherosclerosis. Inhibition of lncRNA-p21 increases proliferation and reduces apoptosis by binding to the mouse double minute 2 (MDM2) protein and preventing degradation of p53. This lncRNA was significantly repressed in carotid and mononuclear cells from patients with atherosclerosis. Furthermore, its inhibition induced the formation of neo-intima in animal models [14]. Moreover, a highly expressed lncRNA was identified by stimulating VSMC with angiotensin II. Lnc-Ang362 was identified as a new transcript linked to miR-221 and miR-222, which have a pivotal role in VSMC proliferation [15].

Recently, the competing endogenous RNA (ceRNA) hypothesis has unveiled another area of paramount importance, namely, the role of lncRNAs as epigenetic actors. MicroRNAs are negative regulators of gene expression, decreasing the stability of target RNAs or limiting their translation. According to ceRNA theory, two or more transcripts can compete for microRNA response elements (MRE); thus, they act as trans-regulatory factors [16]. LncRNA-mediated ceRNA have been associated with cancer development, prognosis [17], and therapeutic response [18]. CeRNA crosstalk has been assessed in cardiovascular disease as well [19], highlighting the relevance of this epigenetic regulatory mechanism in vascular dysfunction in response to diverse stimuli. GAS5 inhibits PDGF–BB-induced VSMC proliferation and migration, partly by acting as a ceRNA of miR-21, providing new evidence that GAS5 may serve as a potential therapeutic target for hypertension. [20]. Meanwhile, in vascular endothelial cells (ECs) treated with oxLDL, lnc-MKI67IP-3 acted as a sponge for let-7e, suppressing its pro-inflammatory effects and thereby upregulating IκBβ expression, forming a positive feedback loop to aggravate inflammation. Thus, this ceRNA axis might play important roles in the inflammatory responses of ECs and development of atherosclerosis [21].

The growing importance of lncRNAs in the pathogenesis of cardiovascular diseases along with their tissue-specific expression motivated us to evaluate the expression profile of these transcripts during the phenotypic switch of VSMCs induced by bare metal stent and PDGF-BB exposition. This will contribute to understanding the mechanisms that rule the change in smooth muscle cell phenotype and represents the first step in a pipeline of genetic biomarker discovery to allow for early diagnosis of post-angioplasty coronary stent restenosis.

## 2. Results

### 2.1. Phenotypic Change Induction by Stent and PDGF-BB

HUASMC grown under different conditions are shown in Figure 1. Differences in growth pattern were observed by inducing differentiation of the contractile phenotype, with elongated, spindle-shaped cells. HUASMC of the stent-induced injury model showed a reticular organization with clear nuclear definition, especially in the vicinity of the stent (Figure 1A). Furthermore, the stent injury model significantly promoted the proliferation of HUASMC (Figure 1B). Contractile cells were quiescent, most in G0-G1 cell cycle phases. Cells entered the S and G2-M phases under stent and PDGF stimuli (Figure 1C).

### 2.2. Stent Injury Model Promotes Cells Migration and the Expression of Genetic Markers of Phenotypic Switch

Cell migration was evaluated by wound healing assay, which revealed that the migratory capacity of HUASMC increased in the stent injury group (Figure 2A,B). Genes associated with a contractile or synthetic/proliferative phenotype are plotted in Figure 2C [3,22]. In the stent-stimulated group, there was lower expression of contractile phenotype-related genes such as smooth muscle α-actin (ACTA2), calponin (CNN1), and collagen type IV (COL4A1), whereas genes associated with the synthetic/proliferative phenotype, such as osteopontine (SPP1), matrix metalloproteinase 1 (MMP1), and fibronectine (FN1), were upregulated.

### 2.3. Stent Injury Model Drives Activation of ncRNA Genes along with mRNA Downregulation

The transcriptomic profile of the stent injury model cells revealed activation and suppression of both coding and non-coding genes (ncRNA) (Figure 3A). The relative abundance of ncRNA was superior to mRNA, representing 65.7 % of the mapped transcripts (Figure 3B). Using a threshold of absolute Fold Change ≥ 2 and an FDR ≤ 0.05, we identified 3517 differentially expressed (DE) mRNA and 2911 DE ncRNA. There was no difference in expression level among DE ncRNA, and DE mRNAs were mostly downregulated (Table 1).

### 2.4. Identification of Differentially Expressed Genes in HUASMC Exposed to PDGF and Stent

Differential expression analysis confirmed substantial differences in mRNA and ncRNA expression between the contractile and stimulated cells. Using more stringent criteria, FDR ≤ 0.01 and absolute log2FC ≥ 2 to declare significance and FPKM > 1 to confirm quantifiable expression, we identified 1099, 509, and 64 differentially expressed mRNA, lncRNA, and microRNA, respectively (Figure 4). Table 2 shows that most of the protein coding genes (72.1%) were silenced during the phenotypic switch of HUASMCs, while lncRNAs were overexpressed in this process (59.3%). MicroRNAs did not show any differences under the experimental conditions.

Table 3 shows the top twenty dysregulated lncRNA during phenotypic switch induced by stent and PDGF. Among the top ten upregulated genes six have not previously been annotated, and there are two antisense transcripts and two long intervening noncoding RNAs (lincRNAs) as well. Meanwhile, in the downregulated group there is only one novel LncRNA along with four antisense, two lincRNAs, and two sense-intronic RNAs.

### 2.5. Gene Enrichment Analysis of DE mRNA and Cis-Target mRNA of lncRNAs

GO analysis of DE mRNA showed significant enrichment for terms related to collagen fibril organization (GO: 0030199), extracellular matrix organization (GO: 0030198), regulation of chemokine-mediated signaling pathway (GO: 0070099), and regulation of smooth muscle cell proliferation (GO: 0048660). Moreover, notable KEGG terms included protein digestion and absorption, IL-17 signaling pathway, AGE-RAGE signaling pathway in diabetic complications, and TGF-beta signaling pathway, among others (Figure 5A,B).

The complementary pairing of lncRNA and mRNA bases as well as target gene prediction of LncRNA was performed using the LncTar tool. LncTar uses the complementary sequence between LncRNA and mRNA to calculate the free energy of the pairing site. Additionally, Cis-target gene prediction analysis was performed; thus, the protein coding genes of the 100K region upstream and downstream of DE lncRNA were screened out as its target genes. Finally, the of Cis and LncTar target genes were combined for subsequent analysis.

There were 1095 target mRNAs identified, 544 for downregulated lncRNAs and 551 for upregulated lncRNAs. GO analysis of lncRNA–mRNA target genes showed enrichment for terms related to wound healing, spreading of epidermal cells (GO:0035313), negative regulation of pathway-restricted SMAD protein phosphorylation (GO:0060394), collagen biosynthetic process (GO:0032964), and blood vessel endothelial cell proliferation involved in sprouting angiogenesis (GO:0002043) (Figure 5C). On the other hand, the main enriched KEGG pathways included extracellular matrix–receptor interaction, fatty acid biosynthesis, and protein digestion and absorption (Figure 5D).

### 2.6. Competing Endogenous RNA (ceRNA) Network Construction

According to the ceRNA theory, a pair of co-expressed lncRNA and mRNA compete for miRNA target sites; therefore, lncRNAs serve as endogenous sponges that target one or more miRNAs, affecting the post-transcriptional regulation of mRNAs. The ceRNA network was constructed with 1241 interactions comprising 9 miRNAs, 39 lncRNAs, and 158 mRNAs; 4 downregulated miRNAs interact with 15 upregulated lncRNAs and 49 upregulated mRNAs (Figure 6A). Meanwhile, 5 upregulated miRNAs interact with 24 downregulated lncRNAs and 109 downregulated mRNAs (Figure 6B).

### 2.7. PPI Network Analysis

We conducted a protein–protein interaction (PPI) network analysis to explore the most significant clusters of DE mRNA among the 150 mRNA in the ceRNA network. The STRING database version 11.5 was used to obtain significant interactions, and the resulting network was visualized in Cytoscape (Figure 7A). The MCC (maximum clique centrality) method from the cytoHubba app in Cytoscape was used to screen for key proteins (Figure 7B). The top upregulated genes were HIF1A, MMP1, and FGF2 while the top downregulated genes were TP53, ELN, and BCL2L11.

### 2.8. Key CeRNA Selection

To identify the most significant lncRNA–miRNA–mRNA ceRNAs, we intersected the ceRNA network (Figure 6) with the genes in the PPI network (Figure 7B). There were 7 mRNA, 7 miRNA, and 27 lncRNA included in the resulting network (Figure 8A). The main lncRNA nodes were selected using the CytoHubba MCC score. Four main lncRNA–miRNA–mRNA subnetworks were identified: overexpressed PVT1 can decoy four miRNAs (miR-27b/miR-145/miR-196a/miR-152) to modulate the expression of six mRNAs (BNIP3/HIF1A/MMP1/FGF2/TGFA/CD274) with a high correlation coefficient (r > 0.89 for all PVT1–mRNA interactions). Upregulated HIF1A-AS2 might sponge miR-27b and miR-196a to de-repress BNIP3 and FGF2, respectively, although with a lower Pearson coefficient (r < 0.7). There were several ceRNA axes comprising downregulated lncRNA ACTA2-AS1 and MIR663AHG: ACTA2-AS1/miR-29a/ELN; ACTA2-AS1/miR-221/BCL2L11 and MIR663AHG/has-let-7a/BCL2L11; MIR663AHG/miR-29a/ELN; MIR663AHG/miR-324/TP53. Pearson correlations were strong, except for BCL2L11 (Figure 8B).

## 3. Discussion

Vascular smooth muscle cell phenotypic switch is a dynamic process. Individual cells within a tissue can be found in intermediate steps between two opposite edges [22]. In our experimental model, HUASMC exposed to bare metal stent and PDGF-BB changed their cytoarchitecture and gained proliferative and migratory capacities.

Santin et al. exposed VSMCs to stent fragments in vitro, resembling the stent restenosis process [23]. Bare metal stent increased PDGF-BB receptor expression in VSMCs; moreover, under this stimulus, VSMCs secreted growth factors, including PDGF-BB. Cells in direct contact with the stent may activate distant cells and themselves in a paracrine and autocrine manner, respectively [24].

PDGF-BB is a potent growth factor with tyrosine kinase activity that activates cell proliferation via ERK [25], promoting the cell cycle from G1 to S phase [26]. In our in vitro model, the contractile phenotype was successfully reverted to a synthetic/proliferative phenotype, allowing the cell cycle to progress to the S and G2-M phases.

Endothelial cells (ECs) reside near VSMCs in an intact vasculature and play a crucial role, interacting with the underlying VSMCs through direct cell–cell contact or through the synthesis and release of mediators into the surrounding medium [27]. In the co-culture system, quiescent ECs produce heparin-like substances which maintain VSMCs in a “contractile” phenotype, while activated ECs stimulate the proliferation of VSMCs [28]. Under quiescent conditions, co-culture with human umbilical VSMCs attenuates the expression of eNOS of human umbilical ECs, which can be inhibited by shear stress [29]. Microvesicle-derived microRNA traffic is another mechanism of paracrine cell–cell communication [30].

With our in vitro model, we tried to replicate the pathological effects of BMS on the blood vessel, and we were able to induce the phenotypic switch of VSMCs from quiescent/contractile to secretor/proliferative. Proliferative cells overexpressed genes related to matrix deposition and organization (MMP1, FN1 and SPP1). Conversely, contractile cells overexpressed the actine organization-related genes ACTA2 and CNN as well as COL4A1, a known component of “healthy” matrix [4,32]. Collagen type 4 induces the binding of serum response factor (SRF) and myocardin to the CArG box of promoters of α-actin [31] and the smooth muscle myosin heavy chain to upregulate transcription of contractile genes. [32]

Cells harboring proliferative phenotype showed global repression of coding genes while overexpressing lncRNAs. LncRNAs can regulate mRNA expression positively or negatively through different mechanisms: (1) epigenetic regulation, which modulates the state of the chromatin; (2) transcriptional regulation, which modulates RNA polymerase II activity; (3) nuclear organization, which maintains nuclear structures; (4) post-transcriptional gene regulation, either through base pairing with mRNAs or by acting as cofactors or competitors of RNA-binding proteins; and (5) acting as ceRNAs by means of sponging miRNAs, thus de-repressing mRNAs [33]. Furthermore, mRNA expression in VSMCs can be regulated by transcriptional factors, DNA methylation, alternative splicing, and microRNA-mediated post-transcriptional regulation [34]. Although the molecular complexity of VSMC plasticity is determined by all of these mechanisms [35], our investigation focused on the ceRNA effect of lncRNAs.

LncRNA annotation of VSMC transcriptomics in pathological states is hampered by their incomplete representation in GENCODE. Bennett et al. described a repertoire of newly discovered lncRNAs in VSMCs stimulated with PDGF-BB, IL-1α, and vascular stiffness. Remarkably, they found an over-representation of novel VSMC-enriched lncRNAs with pathological association [36]. Our discovery pipeline allowed us to describe 289 newly-assembled transcripts, representing 9.8% of dysregulated lncRNAs (Appendix A).

The multistep analysis pipeline carried out here allowed us to identify a limited number of lncRNAs potentially involved in the regulation of the phenotypic switch of VSMCs using the ceRNA model. Several studies have highlighted the role of the lncRNA plasmacytoma variant translocation 1 (PVT1) in cancer and cardiovascular disease [37]. PVT1 was upregulated in human aortic smooth muscle cells (HASMCs) exposed to PDGF-BB in an artery dissection animal model. Moreover, PVT1 inhibition reversed its proliferative phenotype and was associated with upregulation of miR-27b [38], as described in our model. PVT1 promoted switching from contractile to synthetic phenotype in cultured VSMCs stimulated with angiotensin II (Ang II). Furthermore, knockdown of PVT1 reversed these Ang II-induced changes [39].

PVT1 can modulate the expression of the PI3K/AKT, Wnt5a/Ror2, E2F2, and HIF-1A metabolic pathways, acting as ceRNA [40]. In a cardiac fibrosis model, PVT1 enhanced HCN1 expression via sponging miR-145 [41]. Cardiomyocyte hypertrophy and remodeling via OSMR was modulated by PVT1 by relieving miR-196b-induced OSMR suppression in a ceRNA manner [42]. PVT1 was further associated with proliferation and cisplatin resistance in oral squamous cell carcinoma. Mechanistic investigation revealed that PVT1 positively regulates HIF1a expression through its ceRNA activity on miR-194-5p [43]. In gastric cancer cells, PVT1 was describes as acting as a sponge of miRNA-152, upregulating fibroblast growth factor 2 (FGF2) [44], as in one of our proposed ceRNA axes.

PVT1 has not been previously associated with SR; nevertheless, the mechanism underlying VSMC phenotypic switching is common to SR, atherosclerosis [45], artery dissection [38], and aneurism formation [39]. Furthermore, its reliability in serum samples as a biomarker of cardiac damage [46] and cancer [47] makes PVT1 an attractive therapeutic target and a potential biomarker of SR.

Endogenous hypoxia-inducible factor 1α antisense RNA2 (lncRNA HIF1A-AS2) is involved the development and progression of bladder cancer, glioblastoma, breast cancer, and osteosarcoma [48]. HIF1A-AS2 exerts its function in gastric cancer by acting as ceRNA bound to miR-429, modulating the expression of programed cell death ligand 1 (PD-L1) [49]. In lung cancer cells, HIF1A-AS2 was the sponge for miR-153-5p, and miR-153-5p targeted S100A14; thus, HIF1A-AS2 promoted S100A14 expression by regulating miR-153-5p. [50]. In the vascular field, HIF1A-AS2 acts as a decoy for miR-153-3p, activating the HIF-1α pathway in HUVEC cells and promoting angiogenesis in hypoxic conditions [51]. HIF1A-AS1 contributes to ventricular remodeling after myocardial ischemia/reperfusion injury by adsorption of microRNA-204, regulating SOCS2 expression [52]. Moreover, HIF1A-AS2 is involved in VSMC proliferation induced by PDGF-BB via the miR-30e–5p/CCND2 mRNA axis [53].

LncRNA ACTA2 Antisense RNA 1 (ACTA2-AS1) is located at 10q23.31. Although poorly studied, recent work has shown evidence of ACTA2-AS1 dysregulation in several cancers. ACTA2-AS1 was downregulated in human colon adenocarcinoma tissues and colon adenocarcinoma cell lines, and over-expression of ACTA2-AS1 inhibited cell proliferation and colony formation abilities and regulated apoptosis. In addition, a luciferase reporter assay revealed that ACTA2-AS1 interacted with miR-4428, modulating BCL2L11 expression [54]. In liver cancer cells, ACTA2-AS1 was typically downregulated, and knockdown of this lncRNA significantly promoted the proliferation, cell cycle progression, migration, and invasion of LM3 cells by suppressing E-cadherin and caspase 3 expression and increasing N-cadherin, cyclin D1, and MMP expression [55]. ACTA2-AS1 was identified as a downregulated lncRNA in lung adenocarcinoma samples and cells. Functionally, ACTA2-AS1 overexpression restrained cell proliferation and accelerated cell apoptosis in lung adenocarcinoma cell lines. ACTA2-AS1 exerts functions as a ceRNA by serving as a sponge for miRNA-378a-3p and miR-4428, elevating SOX7 expression [56].

LncRNA MIR663AHG, the host gene of mir-663a, has been linked to Turner syndrome [57] and may play a protective roll in intervertebral disc degeneration [58], psoriasis [59], and pancreatic cancer [60]. In our model, MIR663AHG was downregulated during the phenotypic switch, probably leading to de-repressing of hsa-let-7a, miR-324, and miR 29a with downregulation of TP53, ELN, and BCL2L11.

The miR-29-ELN (elastin) axis has been described in VSMC osteoblastic phenotypic transformation. Elastin downregulation promotes calcium deposition on VSMCs, and this phenotype can be reverted with anti-miR-29 [61]. In rat VSMCs stimulated with Ang II, miR-29b was found to bind lncRNA Xist, which can boost ELN levels by absorbing miR-29b. This ceRNA axis has been proven to aggravate apoptosis in human VSMCs [62].

The transcriptomic landscape of VSMC phenotypic switch under conditions resembling stent-induced injury showed dysregulation of a great many transcripts, many of them previously described as being associated with contractile/synthetic balance. Nowadays, bioinformatics analysis of next-generation genetic data combined with in silico approaches have allowed researchers to select potentially relevant targets from among the ocean of potential candidates in order to further explore their biologic relevance [63].

Nowadays, improvement of RNA transfection techniques and development of new biomaterials represent promising therapeutic alternatives [64]. The application of gene expression-modulating stents able to release specific small interfering RNAs (siRNAs) [65], mRNAs [66], microRNAs [67], or lncRNA into the vascular wall may have the potential to both improve regeneration of the vessel wall and inhibit adverse effects. The lncRNA SENCR alleviated the inhibitory effects of rapamycin on HUVECs, and may be useful as a new combinative agent to avoid the disadvantages of mTOR inhibitors in DES implantation [68].

Our in vitro model of stent restenosis has been previously validated; however, there are several limitations: (1) while indirect contact of HUVEC and HUASMCs allows molecule and microvesicle traffic, it impedes cell–cell contact [69]; (2) rheological factors such as shear stress and flow play an important role in vascular cell biology and regulation [29]; (3) alternative strategies such as in vivo animal models [70] and in vitro blood vessel constructs [71] closely resemble real live behavior during stenting; however, to the best of our knowledge there are no precedents for next-generation RNA sequencing of VSMCs co-cultured with HUVECs in a stent-induced injury model combining bare metal stent and PDGF-BB. Thus, the proposed ceRNA axis needs to be validated with functional assays in order to confirm its biologic significance [72].

In summary, overexpression of PVT1 and HIF1A-AS2 were associated with the synthetic/proliferative phenotype, while ACTA2-AS1 and MIR663AHG were protective, resulting in their repression in stent-induced injury cells. PVT1 may act as a ceRNA, de-repressing genes related to the proliferative phenotype, such as HIF1A, MMP1, FGF2, and TGFA, and probably sponging miRNA-145 and miRNA-152, thereby serving as an attractive target to reverse VSMC phenotype switching during the stent restenosis process.

## 4. Materials and Methods

### 4.1. Cell Culture

Human umbilical artery smooth muscle cells (HUASMC) were grown on M231 medium supplemented with Smooth Muscle Growth Supplement (SMGS), 1% streptomycin, and 1% penicillin, at 37 °C in a humidified atmosphere under 5% CO2 conditions. After 24 h, the medium was changed to Smooth Muscle Differentiation Supplement (SMDS). HUASMCs differentiate to the contractile phenotype after ten days in this conditioning medium according to the supplier’s protocol [73].

Human Umbilical Vein Endothelial Cells (HUVEC) were grown on Dulbecco’s modified Eagle’s medium (DMEM) with 10% fetal bovine serum, 1% streptomycin, and 1% penicillin at 37 °C in a humidified atmosphere under 5% CO2. The experiments were performed on cells that had undergone 5–8 passages. Cell lines and reagents were purchased from Thermo Fisher Scientific, Inc. (Waltham, MA, USA).

### 4.2. Stent-Induced Injury Model

Contractile HUASMCs were seeded as a monolayer in 24 well plates. An indirect co-culture system was established, seeding HUVEC on inserts of 0.4 mm pore size with a distance from the membrane of the insert to the bottom of the well of 1.3 mm (SPLInsert™ Hanging). This system allows cells to exchange soluble factors without direct contact [69]. HUASMCs in this condition served as control (contractile phenotype).

Co-cultured cells were then exposed to fragments of bare metal stent (VeriFLEX™ Boston Scientific USA, San Jose, CA, USA). Stent fragments were cut into 6 mm diameter fragments, sterilized with 70% ethanol, and washed three times with Hank’s balanced salt solution. Additionally, cells were treated with PDGF-BB 20 ng/mL (Sigma-Aldrich Co. LLC) (Louis, MO, USA) for 36 h. Under these conditions (Figure 9) HUASMCs developed a synthetic/proliferative phenotype [74].

### 4.3. Cell Migration

Cell migration capability was measured by wound healing assay. When HUASMCs grew to 90% confluency, they were scratched with a sterile 100 μL pipette tip. Cell debris were removed by washing with Hank’s balanced salt solution. Cells were observed after every 12 h under an inverted microscope until confluency was reached. The assay was performed in triplicate [75]. Images were processed with the software Image Processing and Analysis in Java (ImageJ v.1.48). The proportion of covered area was determined for each condition using the plug-in MRI_Wound_Healing_Tool [76].

### 4.4. Cell Cycle Analysis

HUASMCs in both experimental conditions were harvested, centrifuged and pelleted. Ethanol 70% at 4 °C was added drop by drop under continuous agitation. Cells were washed twice with PBS and 2% fetal bovine serum and resuspended in 300 μL medium containing 20 μL of RNasa A (Invitrogen, Waltham, MA, USA) and 2 μL propidium iodide (Sigma Aldrich, Burlington, MA, USA). After mixing, cells were incubated for 30 min in the dark [77]. Cell cycle analysis was performed on a FACs Canto II (Becton-Dickinson, Minato, Tokyo, Japan).

### 4.5. RNA Extraction

Total RNA was extracted from cells using TRIzol reagent (Invitrogen) (Waltham, MA, USA) according to the manufacturer’s protocol. RNA grade Glycogen 1 μg/μL (Thermo Fisher Scientific, Inc., Waltham, MA, USA) was added for enhanced RNA precipitation. RNA purity and quantification were assessed in NanoDrop one and RNA integrity (RIN) was assessed using an Agilent 4200. (Appendix A).

### 4.6. RNAseq Analysis

Ribosomal depleted unstranded libraries were prepared by CD Genomics (Shirley, NY, USA) and sequenced with an Illumina HiSeq X10 PE150 (paired end). High-quality clean data were obtained after a series of quality controls that included (1) removing reads containing adapters; (2) removing reads containing *n* > 10%, where *n* represents the base which cannot be determined; (3) removing reads containing low quality-base representing over 50% of the total base.

Between 55 million and 73 million paired reads were obtained for the samples, allowing a deep coverage. The filtered clean reads were mapped to the reference genome by HISAT2 [78]. (Appendix A).

### 4.7. Gene Expression Quantitative Analysis

Expression levels were measured as Fragments per Kilobase of transcript per Million mapped reads (FPKM). The quantification and gene expression levels of the transcripts were assessed using the Cuffquant and Cuffnorm components of Cufflinks software, using the mapped reads’ positional information on the gene [79].

Differential expression was performed utilizing DESeq2 [80]. We considered a threshold of absolute Fold Change (FC) ≥ 2 and a False Discovery Rate (FDR) < 0.05 to identify significant changes between two conditions. Hierarchical clustering based on the log2(FC) of differentially expressed genes was evaluated with DESeq2 Principal component analysis tool. (Appendix A). Heatmaps were produced in R using the bioconductor package ComplexHeatmap [81].

### 4.8. Filtering of Candidate lncRNAs

To screen for lncRNA within the merged transcripts, we performed basic filtering including five steps:(1)Screening exons, filtering low-expression and low-quality single exon transcripts;(2)Selecting transcripts which were longer than 200 bp and had more than two exons;(3)Screening transcripts with known annotations using Cuffcompare;(4)Transcript expression level filtering by calculating the expression level of each transcript and choosing those with FPKM ≥ 0.5;(5)Coding potential filtering; by combining several mainstream coding potential analysis methods (CNCI [82], CPC [83], and Pfam [84]), it was determined which transcripts had no coding potential (Appendix A).

### 4.9. Prediction of lncRNA Target Genes

The basic principle of Cis target gene prediction is that the function of lncRNA is related to its adjacent protein-coding gene; thus, the protein coding genes of the 100 Kb region upstream and downstream of a lncRNA were screened out as its target gene. Other criteria were based on the complementary pairing of lncRNA and mRNA bases. LncTar uses the complementary sequence between LncRNA and mRNA to calculate the free energy and normalized free energy of the pairing site. Thus, the target gene of the LncRNA is that wich normalized free energy is below the threshold. Finally, the target genes of Cis and LncTar were combined for subsequent analysis [85].

### 4.10. Functional Analysis

Gene ontology (GO; Gene Ontology Consortium, 2000) enrichment analysis of differentially expressed genes was carried out using topGO [86]; Fisher’s exact test was used to calculate *p*-values. The specific genes that are involved in the major metabolic pathways and signal transduction pathways can be determined by Pathway significant enrichment analysis. The Kyoto Encyclopedia of Genes and Genomes (KEGG) is a database resource for understanding the high-level functions and utilities of biological systems. Therefore, we employed the KOBAS software to detect the enrichment of differentially expressed mRNA and lncRNA target genes. The pathways with FDR ≤ 0.05 were defined as the significantly enriched pathways [87].

### 4.11. Construction of lncRNA–miRNa–mRNA Competing Endogenous RNA (ceRNA) Network

We constructed a ceRNA network based on the hypothesis that lncRNAs regulate the activity of mRNAs by sequestering and binding miRNAs, thereby acting as miRNA sponges [88]. All transcripts were filtered according to expression profiles of fold change log2(FC) > 2.5 for lncRNAs and log2(FC) > 1.5 for miRNA and mRNA, with FDR < 0.01. More stringent criteria were used in order to ascertain the biological significance of DE transcripts, as lncRNA’s low abundance and nuclear localization may hamper its ceRNA ability [89]. To construct the ceRNA network, we predicted miRNA–mRNA and lncRNA–miRNA interactions based on miRTarBase [90], miRDB [91], and TargetScan [92]. The resulting matrix was intersected with our sequencing data, and miRNAs with opposite expression levels to the target lncRNAs and mRNAs were selected. Finally, the lncRNA–miRNA–mRNA network was reconstructed and visualized using Cytoscape software v3.7.2 (San Diego, CA, USA).

### 4.12. Protein–Protein Interaction Network

Target mRNAs included in the ceRNA network were analyzed for protein–protein interaction (PPI) with the STRING online search tool, setting a combined score of ≥0.4. Cytoscape v3.7.2 was used to visualize the PPI network, and significant enriched interactions were selected by CytoHubba using maximum clique centrality score (MCC).

## Figures and Tables

**Figure 1 ijms-23-03074-f001:**
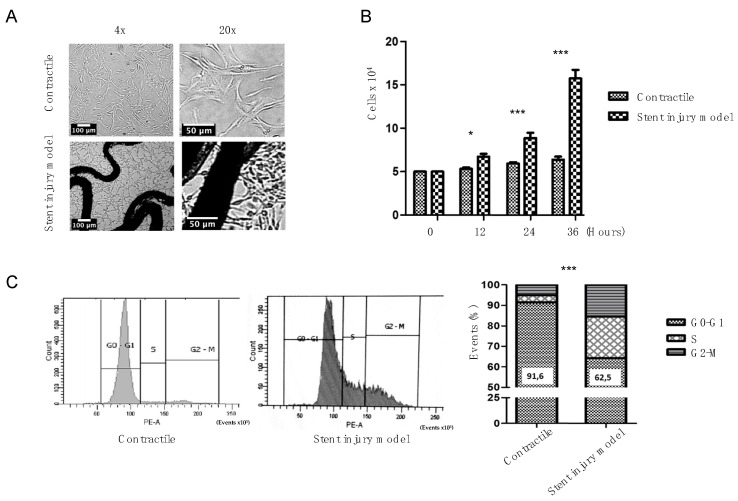
Phenotypic appearance of contractile HUASMC differs from HUASMC under stent injury model conditions. Panel (**A**) shows selected images from both conditions. Stent injury model cells proliferate at a significantly faster rate compared to contractile (**B**). While contractile cells show cell cycle arrest in G0-G1 phase (**C**), HUASMC exposed to stent injury model have a significantly higher proportion of cells in the S and G2-M phases. * *p* < 0.05; *** *p*< 0.001.

**Figure 2 ijms-23-03074-f002:**
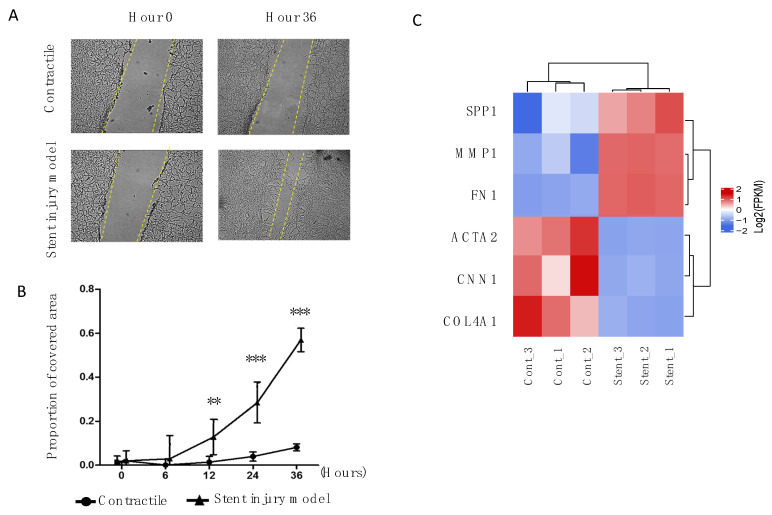
A wound healing assay was performed to compare the migratory capacity in both conditions; selected images of both conditions are depicted in (**A**). Panel (**B**) shows the quantitative expression of this experiment. Contractile cells were mostly quiescent, while stent-induced proliferative cells covered almost 60% of the scratch at 36 h. Panel (**C**) shows the expression of selected genes known to be markers of phenotypic switch; blue and red color intensity accounts for down- and upregulation, respectively. In contractile cells there was an overexpression of ACTA2, CNN1, and COL4A1; the same genes were downregulated in the stent-induced proliferative phenotype, while FN1, MMP1, and SPP1 were uniformly upregulated. Cont_ = Replicates with contractile HUASMC; Stent_ = Replicates with stent-injured HUASMC; ** *p* < 0.01; *** *p* < 0.001.

**Figure 3 ijms-23-03074-f003:**
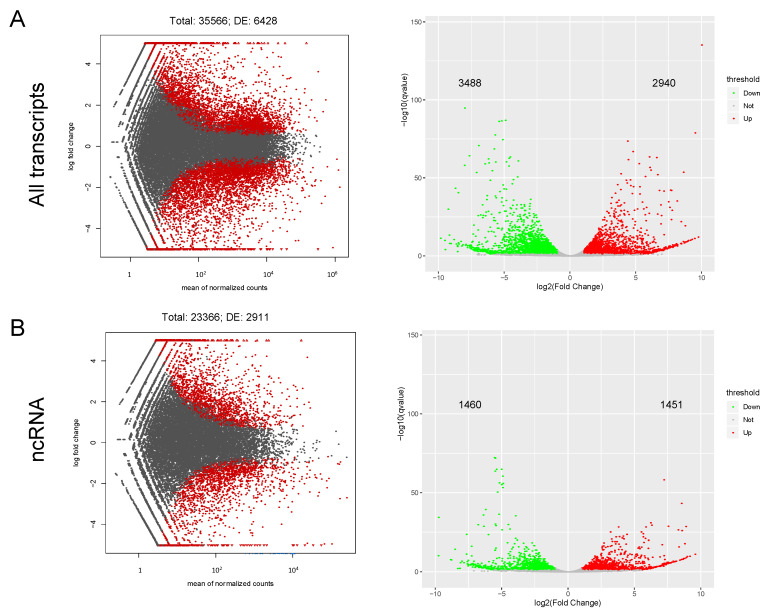
MA scatter plots of sequencing data assessing overall distribution of the two datasets. Expression pattern of all transcripts (**A**) and non-coding transcripts (ncRNA) (**B**). Labeled red dots indicate differential expression (≥two-fold change and FDR ≤ 0.05); volcano plots reflect number, significance, and reliability of differentially expressed transcripts; red dots; and green dots indicate upregulation and downregulation, respectively. The *x*-axis represents the value of log2 Fold change and the *y*-axis represents the adjusted FDR.

**Figure 4 ijms-23-03074-f004:**
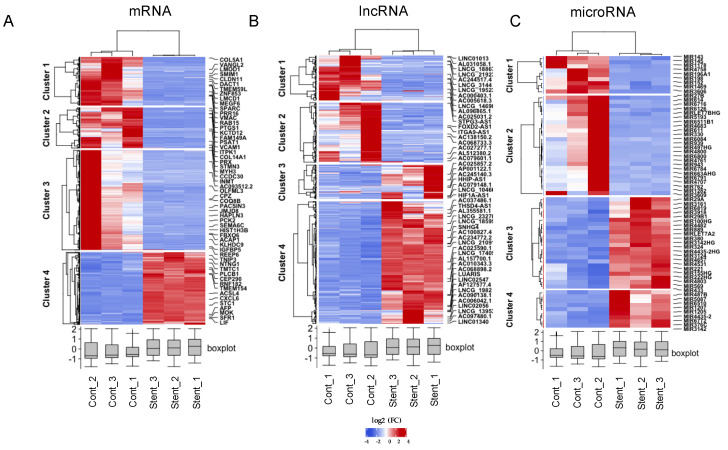
Hierarchical clustering using stringent criteria for DE mRNA (**A**), DE lncRNA (**B**), and DE microRNA (**C**); sample clusters are included above the heatmap and clusters of DE transcripts are noted on the left of each heatmap. Red and blue represent upregulated genes and downregulated genes, respectively. The lower boxplots depict the distribution of each sample gene’s expression. Cont_ = Replicates with contractile HUASMC; Stent_ = Replicates with stent-injured HUASMC; DE: differentially expressed.

**Figure 5 ijms-23-03074-f005:**
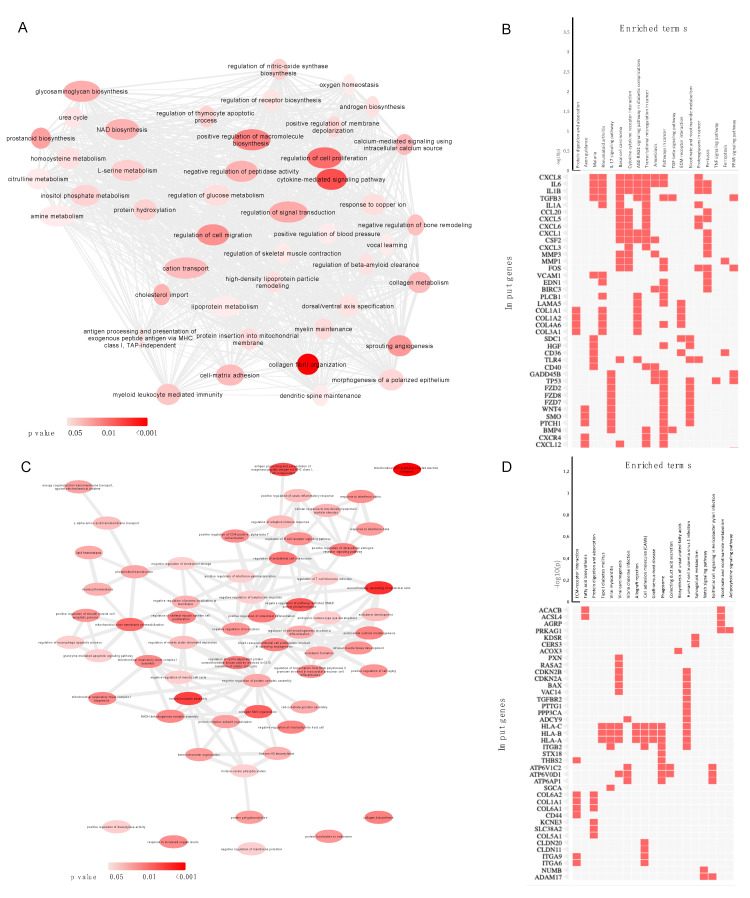
Gene ontology analysis and functional enrichment of KEGG terms for differentially expressed mRNA (panels (**A**,**B**)) and mRNA cis-target of differentially expressed lncRNAs (panels (**C**,**D**)); color intensity depicts significance, while ellipse size represents the number of genes.

**Figure 6 ijms-23-03074-f006:**
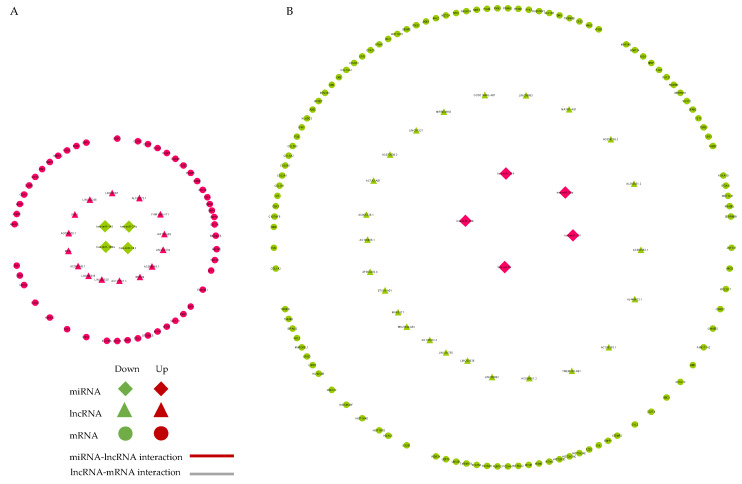
The lncRNA–miRNA–mRNA ceRNA network representation of upregulated (panel (**A**)) and downregulated (panel (**B**)) lncRNAs. Diamonds, triangles, and circles represent miRNAs, lncRNAs, and mRNAs, respectively. Red and green represent upregulated and downregulated RNAs, respectively, in the stent-induced injury cellular model. Red and grey lines indicate miRNA–lncRNA and lncRNA–mRNA interactions, respectively. ceRNA, competitive endogenous RNA; lncRNA, long non-coding RNA; miRNA, microRNA; mRNA, messenger RNA.

**Figure 7 ijms-23-03074-f007:**
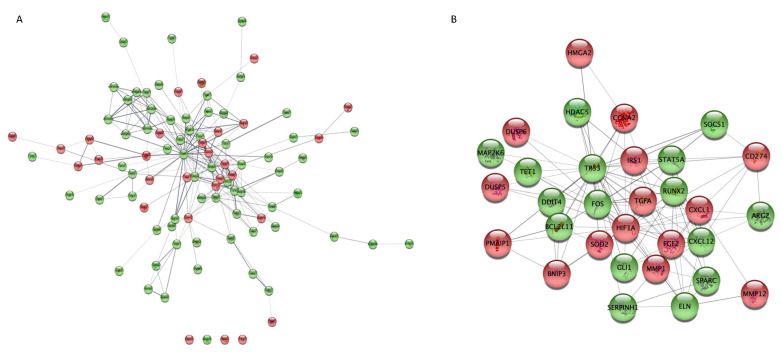
PPI network representation of 150 mRNAs involved in the ceRNA network (panel (**A**)); The network was created by STRING and visualized in Cytoscape. Hub protein network (panel (**B**)) according to CytoHubba MCC coefficient. Red and green represent upregulated and downregulated proteins, respectively; circle size represents MCC score. PPI, protein–protein interaction; ceRNA, competing endogenous RNA.

**Figure 8 ijms-23-03074-f008:**
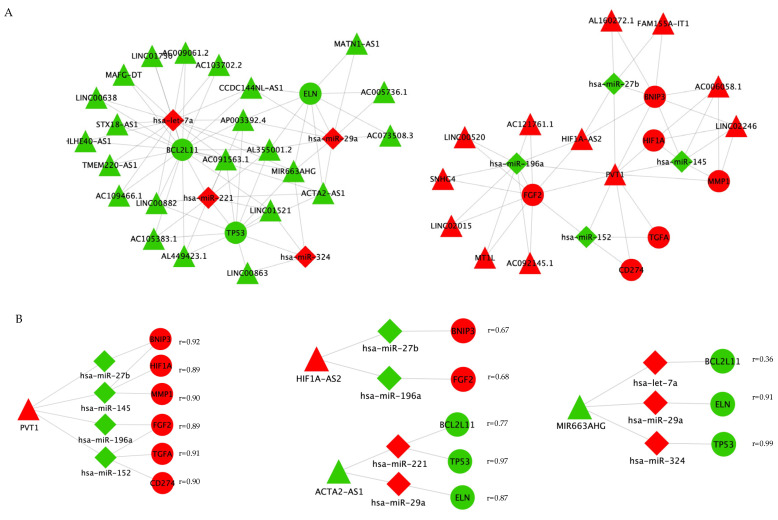
Intersection of ceRNA and hub PPI network in Cytoscape. The network (panel (**A**)) includes 54 nodes (9 mRNA, 8 miRNA, and 27 lncRNA). Four main lncRNAs (PVT1, HIF1A-AS2, ACTA2-AS1, and MIR663AHG) were selected for ceRNA regulatory subnetworks (panel (**B**)). Pearson correlation coefficient for lnrRNA–mRNA interaction (r > 0.5). Red and green represent upregulated and downregulated RNAs, respectively, grey lines indicate interactions. ceRNA, competitive endogenous RNA; lncRNA, long non-coding RNA; miRNA, microRNA; mRNA, messenger RNA.

**Figure 9 ijms-23-03074-f009:**
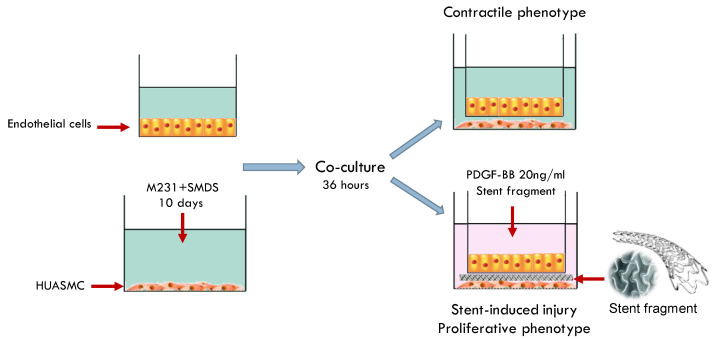
Co-culture system of human umbilical artery smooth muscle cells (HAUSMCs) and endothelial cells with M231 medium plus Smooth Muscle Differentiation Supplement (SMDS). Endothelial cells were seeded on inserts of 0.4 mm pore size in a no-touch manner. For the stent injury model, a piece of bare metal stent was set gently over the HUASMCs. Platelet-derived growth factor (PDGF-BB) 20ng/mL was added to conditioning media.

**Table 1 ijms-23-03074-t001:** Differential expression of transcripts under experimental conditions.

Expression	mRNA		ncRNA	
	N	%	N	%
Downregulated	2028	57.7	1460	50.1
Upregulated	1489	42.3	1451	49.9
Total	3517	100	2911	100

mRNA, messenger RNA, ncRNA, noncoding RNAs. Transcript regulation refers to the expression pattern in the stent injury model over contractile cells. Fisher’s exact test, X^2^ = 36.17; *p* < 0.001.

**Table 2 ijms-23-03074-t002:** Differential expression of significant transcripts under experimental conditions.

Expression	mRNA		lncRNA		microRNA	
	N	%	N	%	N	%
Down regulated	793	72.1	207	40.7	33	51.6
Up regulated	306	27.8	302	59.3	31	48.4
Total	1099	100	509	100	64	100

mRNA, messenger RNA. lncRNA, long noncoding RNA. Transcript regulation refers to the expression pattern in the stent injury model over contractile cells. Fisher’s exact test, X^2^ = 149; *p* < 0.001.

**Table 3 ijms-23-03074-t003:** Top ten upregulated and downregulated lncRNAs in HUASMCs exposed to stent injury.

lncRNA	Expression	log2FC	Location	Type
LNCG_1012	up	11.22	1:70947327–70966066 (+)	novel
AL031429.1	up	10.02	1:70947379–70951493 (−)	antisense
LNCG_20369	up	9.56	5:172802872–172807207 (+)	novel
LNCG_17378	up	8.95	3:91874302–92117628 (−)	novel
AC003092.1	up	8.88	7:94022833–94064723 (+)	lincRNA
LNCG_21417	up	8.86	6:137692987–137705416 (+)	novel
AC079298.1	up	8.77	4:154235980–154237598 (−)	antisense
LNCG_17377	up	8.75	3:91825033–91847766 (−)	novel
AC097480.1	up	8.69	4:28435449–28600275 (+)	lincRNA
LNCG_22261	up	8.56	7:94022772–94068906 (+)	novel
CEBPA-DT	down	−7.19	19:33302857–33305054 (+)	lincRNA
AC103702.2	down	−7.24	17:48635923–48647023 (+)	sense-intronic
AP005131.7	down	−7.29	18:13526078–13526688 (+)	sense-intronic
AC027309.1	down	−7.55	5:172690454–72697720 (−)	antisense
AL109924.2	down	−7.62	6:169034197–169035642 (+)	lincRNA
AC092687.3	down	−7.62	2:10767875–10770058 (−)	antisense
AC023669.1	down	−8.03	7:46261064–46294469 (+)	lincRNA
AP001180.4	down	−8.19	18:10704297–10709599 (−)	antisense
AL445183.1	down	−8.55	1:53069938–53085502 (−)	antisense
LNCG_14690	down	−9.78	2:190677035–190813478 (−)	novel

lncRNA: long noncoding RNA; FC: fold change; Location: Chromosome:base-range (strand).

## Data Availability

Not applicable.

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
