# Peer review of "Long Non-Coding RNAs Might Regulate Phenotypic Switch of Vascular Smooth Muscle Cells Acting as ceRNA: Implications for In-Stent Restenosis"

_ijms, 2022, doi:10.3390/ijms23063074_

Round 1
Reviewer 1 Report
In an experimental model, the authors investigate the changes in the transcriptome architecture during the phenotypic switch of vascular smooth muscle cells (VSMCs). The authors co-cultured VSMCs and endothelial cells and exposed them to bare metal stent and platelet derived growth factor (PDGF-BB). The authors performed RNA-seq to identify the transcriptome changes between contractile and proliferative VSMCs. The topic is significant since VSMCs changes are involved in coronary in-stent restenosis. The authors performed ceRNA (competing endogenous RNA) analysis. They suggest that the PVT1 ceRNA axis may serve as a potential target for in-stent restenosis treatment. The paper is well written, and the topic is important. I have several comments that need to be addressed to improve the readability.
- Section 4.11. states that the cut-off fold change for lncRNA was 2.5, while for miRNA and mRNA was 1.5. Please elaborate on why that difference in stringency.
- Please define in the Table Legends what is in the denominator – the contractile HUASMC or the stent injury model HUASMC expression, so it is clear what up-regulated and down-regulated refer to.
- Under the competing endogenous RNA hypothesis, it would be plausible to think that if there are more lncRNA that can serve as sponges for the miRNA, then less miRNA would be available to suppress the mRNA, thus more mRNA would be up-regulated. Table 2 shows higher number of upregulated lncRNA, and higher number of down-regulated mRNA. Though lncRNAs have other functions beyond serving as molecular sponges, please elaborate as to why the findings in Table 2 might be.
- Figures 5, 6 and 7 (left panel) are unreadable. The axes titles in the rest of the figures also should be larger font for better readability.
- The end of the first paragraph of section 2.5 cites figure 6, when it should be figure 5.
- A few typos, such as:
-“growing factor” should be “growth factor”
- Section 1., beginning of 4th paragraph: “Recent studies in the cardiovascular field points to…” , should be “…point to…”
- Section 3. Discussion, 9th paragraph: “ Furthermore, is reliability in serum samples…” should be “…its reliability…”
Author Response
RESPONSE LETTER
Reviewer 1:
- Section 4.11. states that the cut-off fold change for lncRNA was 2.5, while for miRNA and mRNA was 1.5. Please elaborate on why that difference in stringency.
A: We included the following information: “More stringent criteria were used in order to ascertain biological significance of DE transcripts, as lncRNA low abundance and nuclear localization may hamper its ceRNA ability [90]”
- Please define in the Table Legends what is in the denominator – the contractile HUASMC or the stent injury model HUASMC expression, so it is clear what up-regulated and down-regulated refer to.
A: Table legends were redefined to make clear that regulation degree refers to stent injury model cells: “Transcript regulation refers to the expression pattern in stent injury model over contractile cells”
- Under the competing endogenous RNA hypothesis, it would be plausible to think that if there are more lncRNA that can serve as sponges for the miRNA, then less miRNA would be available to suppress the mRNA, thus more mRNA would be up-regulated. Table 2 shows higher number of upregulated lncRNA, and higher number of down-regulated mRNA. Though lncRNAs have other functions beyond serving as molecular sponges, please elaborate as to why the findings in Table 2 might be.
A: The following information was included:
“Cells harboring proliferative phenotype showed a global repression of coding genes, while overexpressed lncRNAs. LncRNAs can regulate mRNA expression positively or negatively through different mechanisms: (1) epigenetic regulation, modulating the state of the chromatin; (2) transcriptional regulation, modulating RNA polymerase II activity; (3) nuclear organization, to maintain nuclear structures; (4) post-transcriptional gene regulation by base pairing with mRNAs, or acting as cofactors or competitors of RNA-binding proteins; (5) acting as ceRNAs, by means of sponging miRNAs, thus derepressing mRNAs [35]. Furthermore, mRNA expression in VSMC can be regulated by transcription factors, DNA methylation, alternative splicing and microRNA mediated postrascriptional regulation [36]. The molecular complexity of VSMC plasticity is determined by all this mechanisms [37]. Our investigation focused on the ceRNA effect of lncRNAs.
LncRNA annotation of VSMCs transcriptomic in pathological states is hampered by an incomplete representation in GENCODE. Bennett et al described a repertoire of newly discovered lncRNA in VSMCs stimulated with PDGF-BB, IL-1a and vascular stiffness. Remarkably they found an over representation of novel VSMC-Enriched lncRNAs with pathological association [38]. Our discovery pipeline allowed us to describe 289 newly assembled transcripts that represent 9.8% of dysregulated lncRNAs (DataSet S1).”
- Figures 5, 6 and 7 (left panel) are unreadable. The axes titles in the rest of the figures also should be larger font for better readability.
- The end of the first paragraph of section 2.5 cites figure 6, when it should be figure 5.
- A few typos, such as:
A: Figures were improved and reformatted for better readability, and typos errors were extensively corrected.

Reviewer 2 Report
Dear authors,
Thanks for submitting the manuscript "Long non-coding RNAs might regulate phenotypic switch of vascular smooth muscle cells acting as ceRNA: implications for in-stent restenosis" to the journal. It describes activation of VSMCs and RNA family interaction in stent restenosis using bare metal stent injury model in vitro. This manuscript is well written and presented. Results are clear and bring novelty for the biomedical area. Below are some questions and suggestions:
1) Magnification of Figures 1A looks different. Scale is missing
2) Line 167: Are you referent to figure 5 or 4?
4) Line 568: N should be added
3) Interesting if isolated vessels are used in the experiments as well for more realistic approach.
Author Response
Reviewer 2:
- Magnification of Figures 1A looks different. Scale is missing
A: Figure 1A was revised and 20x magnification was set as the correct one. Scale bar was included.
- Line 167: Are you referent to figure 5 or 4?
A: The reference to figure 4 was corrected.
- Line 568: N should be added
A: CONICYT Scholarship (N° 63140032) was included.
- Interesting if isolated vessels are used in the experiments as well for more realistic approach.
A: To address the importance of alternative experimental models the following paragraph was rewrited:
“Our in-vitro model of stent restenosis has been previously validated but have some limitations: 1) Indirect contact of HUVEC and HUASMCs allow molecule and microvesicle traffic but impede cell-cell contacts [71]. 2) Rheologic factors such as shear stress and flow play an important role in vascular cell biology and regulation [30]. 3) Alternative strategies are in vivo animal models [72] and in vitro blood vessels constructs [73], that closely resemble real live behavior during stenting. However, to the best of our knowledge, there are no precedents of next generation RNA sequencing of VSMCs co-cultured with HUVECs in a stent induced injury model combining bare metal stent and PDGF-BB. The proposed ceRNA axis needs to be validated with functional assays to confirm its biologic significance [74].”

Reviewer 3 Report
I have no major correction for this elegant piece of work.
Just in Figure 7 turn to Linotype. The authors might address, in the discussion section therapeutic approaches of regulating mRNA levels by ready to use synthetic constructs see for eg https://www.mdpi.com/1999-4923/13/9/1371, to connect with applications of their biomarkers' findings.
Author Response
Reviewer 3:
- Just in Figure 7 turn to Linotype. The authors might address, in the discussion section therapeutic approaches of regulating mRNA levels by ready to use synthetic constructs see for eg. https://www.mdpi.com/1999-4923/13/9/1371, to connect with applications of their biomarker’s findings.
A: We included the following paragraph:
“Nowadays, improvement of RNA transfection techniques and new biomaterials development show a promising therapeutic alternative [66]. The application of gene expression–modulating stents releasing specific small interfering RNAs (siRNAs) [67], mRNAs [68], microRNA [69] or lncRNA to the vascular wall might have the potential to improve the regeneration of the vessel wall and to inhibit adverse effects. The lncRNA SENCR alleviated the inhibitory effects of rapamycin on HUVECs and may be useful as a new combinative agent to avoid the disadvantages of mTOR inhibitors in DES implantation [70]”
